# STACI: Spatio-Temporal Aleatoric Conformal Inference

**Brandon R. Feng**
North Carolina State University

**David Keetae Park**
Brookhaven National Laboratory

**Xihaier Luo**
Brookhaven National Laboratory

**Arantxa Urdangarin**
University of the Basque Country

**Shinjae Yoo**
Brookhaven National Laboratory

**Brian J. Reich**\*
North Carolina State University

## Abstract

Fitting Gaussian Processes (GPs) provides interpretable aleatoric uncertainty quantification for estimation of spatio-temporal fields. Spatio-temporal deep learning models, while scalable, typically assume a simplistic independent covariance matrix for the response, failing to capture the underlying correlation structure. However, spatio-temporal GPs suffer from issues of scalability and various forms of approximation bias resulting from restrictive assumptions of the covariance kernel function. We propose STACI, a novel framework consisting of a variational Bayesian neural network approximation of non-stationary spatio-temporal GP along with a novel spatio-temporal conformal inference algorithm. STACI is highly scalable, taking advantage of GPU training capabilities for neural network models, and provides statistically valid prediction intervals for uncertainty quantification. STACI outperforms competing GPs and deep methods in accurately approximating spatio-temporal processes and we show it easily scales to datasets with millions of observations.

## 1 Introduction

Accurate estimation of spatio-temporal (ST) fields is a complex task that has relevance in a wide variety of domains [1; 2]. These domains range from, but are not limited to medical imaging[3; 4], remote sensing [5; 6], climate modeling [7; 8] and video quality [9]. Analysis of ST data is difficult as there are often obstructions in the field causing missing data, the surface is rarely smooth across both space and time, and the volume of data requires efficient estimators to be used. These issues have resulted in the need of flexible models providing both accurate prediction of surfaces and uncertainty quantification (UQ) providing interpretability of results.

Gaussian Process (GP) regression is heavily used in spatio-temporal statistics [10; 11; 12]. GPs provide both a flexible predictive surface and precise UQ. Unfortunately, it is not scalable, with the computational cost of likelihood evaluation being cubic in number of locations. Sparse GPs [13; 14] trained from a subset of inducing points, spectral GPs [15] that project the GP into a low-rank spectral domain, and nearest neighbor methods [16; 17] that assume a local dependent structure are three popular classes of approximate GPs that reduce this cost. In much of the approximate STGP literature, the covariance kernel is assumed to have some known stationary and separable form to maintain

---

\*Corresponding author, bjreich@ncsu.edu

39th Conference on Neural Information Processing Systems (NeurIPS 2025).

computational tractability [18; 19; 20]. However, this assumption is often simplistic and fails to hold in complex fields, such as environmental data [21; 22; 23]. Non-stationary ST kernels have been derived, but they can be difficult to compute and can still be mis-specified, requiring more general approaches to modeling non-stationarity [24; 1; 25; 26; 27].

Deep learning methods have risen in popularity, providing scalable and flexible estimation of ST surfaces [28; 29; 30]. Implicit neural representations (INRs) are a class of neural network architectures specifically developed to accurately model the non-linear surface of coordinate-based fields. They typically use periodic activation functions in a multilayer perceptron (MLP) architecture to map spatial ($\mathbb{R}^2$) or ST coordinates ($\mathbb{R}^3$) to the corresponding complex signal domain [31; 32; 33; 34; 30]. However while they are unparalleled in estimation flexibility, they are deterministic functions that do not inherently provide UQ and can overfit to the observed data.

Deep GP models attempt to bridge the gap between neural network flexibility and the interpretability of regular GP models with easily computable measures of variance and subsequent prediction intervals [35; 36]. The covariance kernel is modeled as the output of another GP, and this can be done to an arbitrary depth to approximate any non-stationary kernel. However, similar to regular GPs, there is a trade-off of scalability as even a few layers can create computational bottlenecks. Recently, [37] developed a highly scalable variant of a ST deep GP, projecting separate spatial and temporal GP layers into the spectral domain before concatenating into one unified prediction output. They approximate the spectral process using a probabilistic Bayesian neural network trained using backpropagation, making it highly scalable. However, as this is a deep GP, the covariance parameters lose interpretability as they are used in multiple GP layers. This approximation is most similar to our work that we will now introduce.

The drawback to these approximations is the potential loss of nominal coverage of prediction intervals as the variance estimate is no longer exact. Conformal inference [38; 39; 40] is a model-free approach to calculating valid prediction intervals that achieve frequentist coverage probability by assuming exchangeability of the data. Recent advancements in spatial conformal prediction [41; 42] motivate our extension to the ST setting, guaranteeing validity of our method prediction intervals.

We propose the STACI algorithm[1]: a two-stage approach consisting of combining a Bayesian neural network approximation of a spectral non-stationary STGP with a ST conformal inference algorithm providing valid prediction intervals for accurate UQ. STACI provides:

1. Scalable, flexible interpolation for datasets with millions of ST locations.
2. Ability to both directly compute the underlying correlation structure and interpret estimated covariance parameters, allowing for prior choices to be user-friendly, with the ability to use informative priors if desired.
3. Adaptable prediction interval lengths, clearly showing areas of high and low uncertainty while maintaining desired coverage properties and reducing the impact of both spectral and neural network approximations on UQ accuracy.

## 2 Preliminaries

**Problem Statement.** Assume the ST process $Y(\mathbf{s}, t)$, indexed by location $\mathbf{s} \in [0, 1]^2$ and time $t > 0$, can be decomposed as
$$Y(\mathbf{s}, t) = \mu(\mathbf{s}, t) + Z(\mathbf{s}, t) + \epsilon(\mathbf{s}, t) \tag{1}$$
for mean function $\mu$, mean-zero GP $Z$ and noise $\epsilon(\mathbf{s}, t) \overset{iid}{\sim} \mathcal{N}(0, \tau^2)$. A common model [e.g., 10; 43] for the covariance of $Z$ is the stationary Matérn function
$$\mathrm{Cov}[Z(\mathbf{s}, t), Z(\mathbf{s}', t')] = \sigma^2 \frac{2^{1-\nu}}{\Gamma(\nu)} \left(\sqrt{2\nu}d\right)^\nu K_\nu \left(\sqrt{2\nu}d\right), \tag{2}$$
for $d^2 = ||\mathbf{s} - \mathbf{s}'||^2/\rho_s^2 + (t - t')^2/\rho_t^2$ and Bessel function $K_\nu$. The covariance is defined by the variance $\sigma^2$, spatial range $\rho_s$, temporal range $\rho_t$ and smoothness parameter $\nu$. For simplicity, assume the observed data has already been centered and scaled with the mean and variance estimated separately.

---

[1]Code and data: https://github.com/bf5124/STACI

**Challenges.** Applying this GP framework (1)-(2) directly to large, complex real-world ST datasets faces several significant hurdles:

1. **Computational Scalability:** Exact GP inference requires operations on the $N \times N$ covariance matrix (where $N$ is the number of observations), entailing $\mathcal{O}(N^3)$ computational complexity and $\mathcal{O}(N^2)$ memory requirements. This quickly becomes prohibitive for datasets with thousands, let alone millions, of points [12].

2. **Non-stationarity:** The stationarity assumption, implying a covariance structure dependent only on ST lag $(\mathbf{h}_s, h_t)$, is often unrealistic. Environmental heterogeneities, varying physical dynamics, or boundary effects can cause the dependence structure (e.g., correlation range, variance) to change across space and time [44; 11].

3. **UQ:** While various approximation techniques are employed to address scalability (C1) and non-stationarity (C2), these approximations often break the theoretical guarantees of the exact GP model. Consequently, the resulting predictive uncertainties (e.g., variances, prediction intervals) may lack statistical validity, potentially failing to achieve the desired nominal coverage probability [10; 38].

**Spectral Methods for C1.** To overcome the computational bottleneck (C1), a prominent class of methods leverages the spectral representation of stationary GPs [10; 12]. Bochner's theorem guarantees that any stationary covariance function $C(\mathbf{h}_s, h_t) = \mathrm{Cov}[Z(\mathbf{s}, t), Z(\mathbf{s} + \mathbf{h}_s, t + h_t)]$ is the Fourier transform of a spectral measure. For the Matérn and many other processes, this measure has a density $\sigma^2 f(\boldsymbol{\omega})$, where $f(\boldsymbol{\omega})$ is the normalized spectral density ($\int f(\boldsymbol{\omega})d\boldsymbol{\omega} = 1$) and $\sigma^2 = C(\mathbf{0}, 0)$ is the process variance. The covariance is recovered from the spectral density via the Wiener-Khinchin theorem:

$$C(\mathbf{h}_s, h_t) = \sigma^2 \int_{\mathbb{R}^3} \cos(\boldsymbol{\omega}_s^T \mathbf{h}_s + \omega_t h_t) f(\boldsymbol{\omega}) \, d\boldsymbol{\omega}, \tag{3}$$

where $\boldsymbol{\omega} = (\boldsymbol{\omega}_s, \omega_t)$ represents the ST frequencies. The spectral density $f(\boldsymbol{\omega})$ corresponding to the Matérn covariance (2) is known to have the form of a multivariate Student's t-distribution [10]. This spectral view motivates computationally efficient approximations based on sampling frequencies $\boldsymbol{\omega}$ from $f(\boldsymbol{\omega})$, as detailed next. This spectral view motivates computationally efficient approximations using $J$ random basis functions [45], such as Bayesian Random Fourier Features (BRFF) [46]. The GP, $Z$. is approximated as:

$$Z_J(\mathbf{s}, t) \approx \sum_{j=1}^{J} \left[ \cos(\boldsymbol{\omega}_{s,j}^T \mathbf{s} + \omega_{t,j} t) a_j + \sin(\boldsymbol{\omega}_{s,j}^T \mathbf{s} + \omega_{t,j} t) b_j \right]. \tag{4}$$

In the fully Bayesian BRFF approach, both frequencies and amplitudes are treated as random variables. Priors are chosen such that the resulting process $Z_J$ approximates $Z$: frequencies are sampled $(\boldsymbol{\omega}_{s,j}, \omega_{t,j}) \stackrel{iid}{\sim} f(\boldsymbol{\omega})$ (the Matérn spectral density), and amplitudes are $a_j, b_j \stackrel{iid}{\sim} \mathcal{N}(0, \sigma^2/J)$. Miller and Reich [46] demonstrated the computational efficiency and predictive performance of this using a fully Bayesian approach in a spatial setting. However, their approach is unable to scale to large numbers of spatial, or ST observations, requiring a more computationally efficient method of utilizing the BRFF approximation.

**Dimension Expansion for C2.** To relax the often unrealistic stationarity assumption, dimension expansion methods propose that a non-stationary process $Z(\mathbf{s}, t)$ can be viewed as stationary in an augmented space $[\mathbf{s}, t, \mathbf{L}(\mathbf{s}, t)]$ [47; 44]. Here, $\mathbf{L}(\mathbf{s}, t) = [L_1(\mathbf{s}, t), ..., L_p(\mathbf{s}, t)]$ is a mapping to a $p$-dimensional latent space, representing unobserved factors (e.g., local environmental conditions, dynamic regimes) that modulate the covariance structure. Given $\mathbf{L}$, stationarity is recovered, and a stationary kernel like Matérn (2) can be used with a modified distance incorporating the latent variables:

$$d^2 = ||\mathbf{s} - \mathbf{s}'||^2 / \rho_s^2 + (t - t)^2 / \rho_t^2 + \sum_{j=1}^{p} [L(\mathbf{s}, t) - L(\mathbf{s}', t')]^2 / \rho_j^2. \tag{5}$$

The latent dimension $p$ and the processes $\mathbf{L}(\mathbf{s}, t)$ themselves are typically unknown and must be inferred from the data, adding complexity to the modeling task.

**Conformal Inference for C3.** Addressing the challenge of obtaining statistically valid uncertainty estimates, especially when using approximations for scalability or non-stationarity, can be achieved

using conformal inference [38; 48]. Consider observations $\mathbf{W}_1, \ldots, \mathbf{W}_n$, where each $\mathbf{W}_i = (Y_i, \mathbf{X}_i)$ consists of a response $Y_i \in \mathbb{R}$ and covariates $\mathbf{X}_i \in \mathbb{R}^d$. The core assumption of conformal inference is that the sequence $\mathbf{W}_1, \ldots, \mathbf{W}_n, \mathbf{W}_{n+1}$ (including a new, unseen test point) is exchangeable, meaning their joint distribution is invariant under permutation. Conformal inference operates using a non-conformity measure $\Delta$, where $\delta_i = \Delta(\mathbf{W}_i, \mathbf{W}_{-i})$ quantifies how dissimilar $\mathbf{W}_i$ is from the set $\mathbf{W}_{-i} = \{\mathbf{W}_1, \ldots, \mathbf{W}_{n+1}\} \setminus \{\mathbf{W}_i\}$. The specific function $\Delta$ is user-defined; common choices involve residuals from a fitted model (e.g., $\Delta(\mathbf{W}_i, \mathbf{W}_{-i}) = |Y_i - \hat{\mu}_{-i}(\mathbf{X}_i)|$, where $\hat{\mu}_{-i}$ is fitted on $\mathbf{W}_{-i}$). To construct a prediction interval for $Y_{n+1}$ given $\mathbf{X}_{n+1}$ and training data $\{\mathbf{W}_i\}_{i=1}^n$, conformal considers hypothetical values $y$ for $Y_{n+1}$. For each $y$, it computes the non-conformity score $\delta_i(y) = \Delta(\mathbf{W}_i, \mathbf{W}_{-i})$. The plausibility of $y$ is measured by its conformal p-value:

$$p(y) = \frac{1}{n+1} \left( \sum_{i=1}^{n+1} 1\{\delta_i \geq \delta_{n+1}\} \right), \quad (6)$$

where $\delta_i$ for $i \leq n$ are typically computed using leave-one-out retraining or, more efficiently, using residuals on a held-out calibration set (split conformal inference). The $100(1-\alpha)\%$ prediction interval for $Y_{n+1}$ comprises all values $y$ deemed sufficiently plausible:

$$\Gamma_{n+1}^\alpha = \{y \in \mathbb{R} : p(y) > \alpha\}. \quad (7)$$

Under only the exchangeability assumption, this interval guarantees $P(Y_{n+1} \in \Gamma_{n+1}^\alpha) \geq 1 - \alpha$ [38]. While spatial variants of conformal inference have been developed [41; 42], it has not yet been well explored in the ST setting. The challenges remaining in C1, C2 and C3 motivate our STACI methodology outlined in Figure 1.

## 3  Methodology

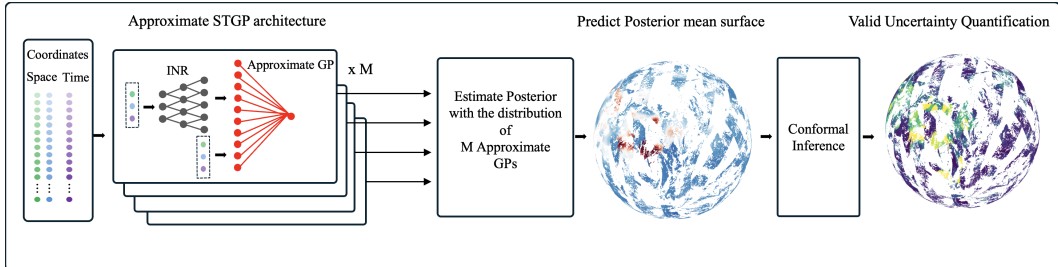

Figure 1: STACI algorithm pipeline. Data from coordinates across space and time are first fed into our approximate spatio-temporal GP architecture and trained using SVGD. A conformal inference step is then fit to provide customized prediction intervals for valid uncertainty quantification.

### 3.1  Bayesian Neural Network Architecture

The greatest drawbacks to the BRFF approximation in (4) are its reliance on assuming a stationary GP kernel, as well as limited scalability stemming from using full MCMC samplers for the frequencies and covariance parameters.

To resolve the first issue, we model $Z$ in (1) using the non-stationary dimension-expansion covariance function in (5). Given latent field $\mathbf{L}$, the spectral approximation is

$$Z(\mathbf{s}, t) = \sum_{j=1}^{J} \cos \left[ \boldsymbol{\omega}_{s,j}^T \mathbf{s} + \omega_{t,j} t + \boldsymbol{\omega}_{L,j}^T \mathbf{L}(\mathbf{s}, t) \right] a_j + \sin \left[ \boldsymbol{\omega}_{s,j}^T \mathbf{s} + \omega_{t,j} t + \boldsymbol{\omega}_{L,j}^T \mathbf{L}(\mathbf{s}, t) \right] b_j, \quad (8)$$

with frequencies $\boldsymbol{\omega}_j = (\boldsymbol{\omega}_{s,j}^T, \omega_{t,j}, \boldsymbol{\omega}_{L,j}^T)$. To approximate a Matérn covariance, we model the frequencies with a multivariate t distribution with $\nu$ degrees of freedom, location zero, and diagonal scale matrix $D$ with diagonal elements $\{\rho_s, \rho_s, \rho_t, \rho_{l,1}, ..., \rho_{l,J}\}$, i.e., $\boldsymbol{\omega}_j \overset{iid}{\sim} MVT_\nu(0, D)$. The amplitudes are modeled as $a_j, b_j \overset{iid}{\sim} \text{Normal}(0, \sigma^2/J)$. For computational simplicity and identifiability concerns, we set $\rho_{l,1} = ... = \rho_{l,J} = \rho_l$. Extending [46] to the nonstationary spatiotemporal case, the following theorem justifies the spectral approximation to the covariance function. The proof is included in Appendix A.1.

**Theorem 1** *The prior mean of the spatiotemporal covariance function of the discrete process in (8) equals the Matérn correlation with distance defined as in (5) for all $J$, and the point-wise prior variance decreases at rate $J$.*

Latent field $\mathbf{L}$ is modeled by an INR with three dimensional input, $(\mathbf{s}, t)$, and $p-$dimensional output, $L_1(\mathbf{s}, t), ..., L_p(\mathbf{s}, t)$. This neural network model for the latent space provides both flexibility and computational scalability for estimating the nonstationary covariance function.

We note that the form of (1), (8) closely resembles a single hidden layer MLP where the first layer is passed through cosine and sine transforms with the bias terms removed and the hidden layer dimension is $J$. Turning this into a Bayesian MLP with priors set to the aforementioned distributions related to (8), this is exactly the form of the BRFF. We then utilize a skip connection, appending the original location set $\mathbf{s}$ and times $\mathbf{t}$ to $\mathbf{L}(\mathbf{s}, t)$ resulting in the final architecture to estimate $Z(\mathbf{s}, t)$. We place relatively uninformative priors on the covariance parameters and utilize the Stein variational gradient descent (SVGD) algorithm developed by [49] to efficiently train our model and obtain variational posterior distributions.

### 3.1.1 Posterior Estimation

We use the SVGD variational inference (VI) framework to approximate the posterior distributions of our neural network model. For some parameter set $\boldsymbol{\theta} = \{\theta_1, ..., \theta_p\} \in \mathbb{R}^p$ and data $\mathbf{Y}$, VI approximates a target posterior, $p(\boldsymbol{\theta}|\mathbf{Y}) = \tilde{p}(\boldsymbol{\theta}|\mathbf{Y})$, using a simpler distribution $q(\boldsymbol{\theta})$, found in a predefined family $Q = \{q(\boldsymbol{\theta})\}$ by minimizing the KL divergence

$$q^*(\boldsymbol{\theta}) = \arg\min_{q \in Q}\{KL(q||p)\mathbb{E}_q[\log q(\boldsymbol{\theta})] - \mathbb{E}_q[\log \tilde{p}(\boldsymbol{\theta}|\mathbf{Y})]\}. \tag{9}$$

Here, $\tilde{p}(\boldsymbol{\theta}|\mathbf{Y})$ is the un-normalized posterior distribution. VI turns posterior estimation into an optimization problem, allowing for greater scalability than sampling-based methods of posterior estimation [50; 51; 52]. However, there can be a trade-off in approximation accuracy. For example, pre-specifying the variational family can result in under-estimation of the posterior variance [52].

SVGD is a VI algorithm that does not require specifying a variational posterior a priori, making it a more generalized approach. For our parameter set $\boldsymbol{\theta} = \{\theta_1, ..., \theta_p\}$, SVGD initializes a set of $M$ independent particles (copies), $\{\boldsymbol{\theta}\} = \boldsymbol{\theta}_1, ..., \boldsymbol{\theta}_M$, that will be trained to approximate the posterior distribution through minimizing the KL divergence between these copies and the target posterior. Given data $\mathbf{Y}$, priors $\pi(\theta_1), ..., \pi(\theta_p)$, joint prior $\pi(\boldsymbol{\theta}) = \prod_{i=1}^{p} \pi(\theta_i)$, joint likelihood $L(\mathbf{Y}|\boldsymbol{\theta})$ and kernel function $\kappa(.,.)$, for each $\boldsymbol{\theta}_i \in \{\boldsymbol{\theta}\}$, the update at iteration $\ell$ with step-size $\epsilon_\ell$ is

$$\boldsymbol{\theta}_i^{\ell+1} = \boldsymbol{\theta}_i^\ell + \epsilon_\ell \phi(\boldsymbol{\theta}_i^\ell), \tag{10}$$

with smooth optimal perturbation

$$\phi(\boldsymbol{\theta}_i^\ell) = \frac{1}{M} \sum_{j=1}^{M} \{\kappa(\boldsymbol{\theta}_j^\ell, \boldsymbol{\theta}_i^\ell)\nabla_{\boldsymbol{\theta}_j^\ell}(\log \pi(\boldsymbol{\theta}_j^\ell) + \log L(\mathbf{Y}|\boldsymbol{\theta}_j^\ell)) + \nabla_{\boldsymbol{\theta}_j^\ell}\kappa(\boldsymbol{\theta}_j^\ell, \boldsymbol{\theta}_i^\ell)\}. \tag{11}$$

As $M \to \infty$, the distribution of $\{\boldsymbol{\theta}\}$ approaches the variational posterior $q(\boldsymbol{\theta})$. In (11), the first term draws particles towards high probability areas of posterior $p(\boldsymbol{\theta}|\mathbf{Y})$ while the second term is a repulsive force that discourages particles from grouping in local modes of the posterior.

### 3.1.2 Neural Network Uncertainty Quantification

Using the SVGD training method gives $M$ trained neural networks $f(., \boldsymbol{\theta}_1), ..., f(., \boldsymbol{\theta}_M) = f_1(.), ...., f_M(.)$. Thus for observation $i$, the MAP estimate of $Y(\mathbf{s}_i, t_i)$ is

$$E[Y(\mathbf{s}_i, t_i)] = E[Z(\mathbf{s}_i, t_i) + \epsilon(\mathbf{s}_i, t_i)] = \frac{1}{M} \sum_{j=1}^{M} f_j(\mathbf{x}_i) \tag{12}$$

with marginal variance

$$V[Y(\mathbf{s}_i, t_i)] = V[Z(\mathbf{s}_i, t_i) + \epsilon(\mathbf{s}_i, t_i)] = E[Z(\mathbf{s}_i, t_i)^2] - E[Z(\mathbf{s}_i, t_i)]^2 + \tau^2. \tag{13}$$

We use the MAP estimate from the trained neural networks, $\hat{\tau}^2$, to estimate the nugget effect. The nugget variance is the random/measurement error term—the aleatoric uncertainty. Given desired type-1 error $\alpha$, letting the mean estimate (12) be $\hat{Y}_i$ and variance (3.1.2) be $\hat{\sigma}_i^2$, we are able to construct $100(1-\alpha)\%$ coverage level credible interval (CI):

$$C_i = (L_i, U_i) = \hat{Y}_i \pm \Phi_{(1-\alpha)/2}\hat{\sigma}_i, \tag{14}$$

where $\Phi_{(1-\alpha)/2}$ is the value of the standard normal cumulative distribution function (CDF) at the $(1-\alpha)/2$ percentile. However, to this point, we have made multiple approximations. First, is the approximation of the full STGP process with our trained Bayesian neural network. Second, is the approximation of the aleatoric nugget variance, $\tau^2$. Thus, our estimated CI cannot be considered statistically valid and we could have over/under estimation of the true variance. The second conformal inference part of STACI serves to correct this issue.

## 3.2 Spatio-temporal Conformal Prediction

We extend the local spatial conformal prediction algorithm of [41] to a ST setting to provide valid prediction intervals for our approximate GP model. To simplify notation, for observation $i$ denote the response as $Y_i = Y(\mathbf{s}_i, t_i)$, the coordinates as $\mathbf{X}_i = (\mathbf{s}_i, t_i)$ and the pair as $\mathbf{W}_i = (Y_i, \mathbf{X}_i)$. The Bayesian computations in the previous section provide fitted values $\hat{Y}_i$ and working standard errors $\hat{\sigma}_i$. As the data is non-stationary over space and time, it is unreasonable to assume exchangeability between all $n$ observations as in Section 2. Thus, the first step is to identify a local set of $K$ approximately exchangeable neighbors across space and time. The neighbors for prediction site $\mathbf{X}_{n+1}$ are selected as the $K$ closest training observations based on squared distance

$$\left[\|\mathbf{s}_i - \mathbf{s}_{n+1}\|/\hat{\rho}_s\right]^2 + \left[|t_i - t_{n+1}|/\hat{\rho}_t\right]^2, \tag{15}$$

where the estimates of spatial and temporal range, $\hat{\rho}_s$ and $\hat{\rho}_t$, are given by the neural network in the first step of STACI. We denote the indices of $K$ neighbors and the prediction location $n+1$ as $\mathcal{N}_{n+1} \subset \{1, ..., n+1\}$.

If we provisionally set $Y_{n+1} = y_{n+1}$ and use discrepancy measure $\delta_j = |y_j - \hat{Y}_j|/\hat{\sigma}_j$, the plausibility is

$$p(y_{n+1}) = \frac{1}{K+1} \sum_{j \in \mathcal{N}_{n+1}} 1\{\delta_j \geq \delta_{n+1}\}. \tag{16}$$

To identify the prediction interval, we search over the range of $Y_i$ among the $K$ neighbors and take the interval as the set of $y_{n+1}$ with plausibility at least $\alpha$. Combining Sections 3.1 and 3.2, the full STACI algorithm provides a scalable non-stationary STGP approximation with valid $100(1-\alpha)\%$ prediction intervals during ST interpolation.

# 4 Application

## 4.1 Setup

### 4.1.1 Data Description

We evaluate STACI's performance on two ST datasets: one synthetic and one real. The first **synthetic** dataset is simulated mean sea surface height (MSS) data of the Arctic sea [37] based on historical satellite data from 3 different tracks. The data spans 10 days from March 1st to March 10th 2020 and has 1,158,505 total datapoints. The second **real** dataset is Aerosol Optical Depth (AOD) data captured using the Moderate Resolution Imaging Spectroradiometer (MODIS) on NASA's Terra satellite [53]. The data is spread on a $1400 \times 720$ grid spanning the Earth's surface and we use daily data from March 2025, equating to 3,189,641 total observations.

The MSS dataset is randomly split into a 80% train, 10% validation, 10% test split. As most observations are seen on each day, this setting tests ST interpolation with small artifacts such as cloud cover creating patches of missing data. For the AOD dataset, we sample 10% of observations randomly per day to comprise the training set. The validation set is all observations over the first 6 days while the test set is all observations on the 20th day. This mimics a task of full reconstruction

over a field when there are limited sensors providing the ground truth data. We note that even the full AOD field is sparse, adding further difficulty to the reconstruction task. For both datasets, the spatial coordinates are scaled to a $[0, 1] \times [0, 1]$ grid and the response is normalized to the training set mean and standard deviation.

### 4.1.2 Model Settings

We test STACI with 3 different state-of-the-art INR architectures to estimate the latent space: Residual Multilayer Perceptron (ResMLP) [34], Fourier Feature Network with Positional encoding (FFNP) and FFN with Gaussian encoding (FFNG) [32]. Each INR backbone has 5 layers with a layer width of 1,024. We set J = 5,000 for the final hidden layer width of STACI, representing the number of random fourier features. We then use $M = 10$ network copies to train using SVGD for the initial Bayesian UQ. Finally, we determine $D$ through cross validation on a random subset of 100 training locations for the number of neighbors to use for conformal inference.

As we are an approximate GP, we compare our algorithm to state-of-the-art scalable GP methods that provide UQ. The first method is sparse variational GP (SVGP) from [14], representing a case of using an approximate stationary GP. The second method is doubly stochastic deep GP from [35], representing an alternate method of estimating a non-stationary GP. The third method is GPSat from [27], modeling the full non-stationary field as a mixture of local, stationary GPs. The final method is deep random features (DRF) from [37], representing a similar algorithm of turning the spectral representation of a GP into a Bayesian deep neural network. SVGP and DeepGP are trained through variational inference, while DRF is trained through Bayesian optimization. DeepGP is set with 4 total layers with layer width 9 and trained with 10 models. GPSat is initialized with 1,225 expert locations across the spatio-temporal domain. DRF is set with 5 hidden layers of width 1,024 and bottleneck layers of width 128 and trained with 10 models. All models are trained on NVIDIA A-100 GPUs for 15 epochs (optimization iterations for DRF) with batch size 1,024. Conformal prediction interval calculation is parallelized over 4 NVIDIA A-100 GPUs.

### 4.1.3 Performance Metrics

We measure the performance of the algorithms in terms of both estimation and UQ quality on the test set of both datasets. For estimation quality, we use root mean square error (RMSE), negative Gaussian log likelihood (NLL). For UQ quality, we use the continuous ranked probability score (CRPS) metric and provide coverage of prediction intervals, interval score and interval width based on $\alpha = 0.05$. Finally, we track time per training epoch to compare computational efficiency. The time for DRF is the time for one optimization iteration. Note that GPSat time is the total time to fit across all expert locations. We provide both Bayesian and conformal UQ performance for STACI. The conformal time represents time needed for the entire conformal step. For RMSE, NLL, CRPS, interval score and interval width, a lower value is better. For coverage, the value closest to 0.95 is deemed the best as estimators providing both over-coverage and under-coverage are considered to be inefficient.

### 4.2 Main Results

Table 1 shows results for the MSS dataset. We see that STACI with the FFNP latent model has the lowest RMSE and NLL, indicating accurate estimation of the surface. We also see this the lowest interval score and interval width of the non-conformal methods while maintaining desired coverage. The conformal interval score is at at worst half that of its competitors, while also being under half as wide as most others. This indicates the individualized interval widths provide much more efficient UQ while achieving the desired coverage level. Total fit time for conformal is also manageable, taking under 8 minutes for the FFN latent models and around 10 minutes for the ResMLP model. We see that SVGP performs much worse than its counterparts in this setting, showcasing the need for inclusion of non-stationarity. GPSat provides comparable estimation metrics to STACI, however the total runtime is over 3 times longer.

Table 2 shows results for AOD dataset. STACI with the FFNP latent model again has the lowest test set RMSE and interval UQ metrics. The conformal addition lowered the interval score by over half, showing the efficiency of this UQ method despite the difficult training and data setting. GPSat had better NLL and near-identical CRPS, albeit again with higher runtime. We note that Deep RF had

Table 1: **Comparisons on the MSS dataset**. Best performances are highlighted in **bold** and underlining for top and second-best methods, respectively.

| Model | RMSE | NLL | CRPS | Coverage | Interval Score | Interval Width | Time (s) |
|---|---|---|---|---|---|---|---|
| STACI-ResMLP (Bayes) | 0.422 | -0.363 | 0.221 | 0.948 | 2.509 | 1.683 | 132 |
| STACI-ResMLP (Conf.) | NA | NA | NA | 0.958 | **0.514** | **0.500** | 652 |
| STACI-FFNP (Bayes) | **0.161** | **-1.331** | **0.086** | **0.951** | 0.923 | 0.663 | 116 |
| STACI-FFNP (Conf.) | NA | NA | NA | 0.958 | **0.514** | **0.500** | 446 |
| STACI-FFNG (Bayes) | 0.247 | -0.907 | 0.129 | 0.948 | 1.458 | 0.992 | 105 |
| STACI-FFNG (Conf.) | NA | NA | NA | 0.958 | **0.514** | **0.500** | 431 |
| Deep RF | 0.203 | -1.112 | 0.106 | 0.970 | 1.172 | 0.941 | 83 |
| Deep GP | 0.277 | -0.782 | 0.145 | 0.957 | 1.633 | 1.171 | 138 |
| GPSat | 0.200 | -1.188 | 0.102 | 0.974 | 1.140 | 0.939 | 7020 |
| SVGP | 0.447 | -0.304 | 0.236 | **0.951** | 2.571 | 1.844 | **39** |

difficulties with this high data sparsity setting and required fixing the nugget variance parameter to achieve convergence.

Table 2: **Comparisons on the AOD dataset**. Best performances are highlighted in **bold** and underlining for top and second-best methods, respectively.

| Model | RMSE | NLL | CRPS | Coverage | Interval Score | Interval Width | Time (s) |
|---|---|---|---|---|---|---|---|
| STACI-ResMLP (Bayes) | 0.720 | 0.361 | 0.400 | 0.916 | 5.506 | 2.656 | 48 |
| STACI-ResMLP (Conf.) | NA | NA | NA | 0.948 | **1.850** | **1.555** | 598 |
| STACI-FFNP (Bayes) | **0.560** | 0.042 | 0.295 | 0.944 | 3.944 | 2.148 | 39 |
| STACI-FFNP (Conf.) | NA | NA | NA | 0.948 | **1.850** | **1.555** | 410 |
| STACI-FFNG (Bayes) | 0.675 | 0.275 | 0.372 | 0.929 | 5.098 | 2.542 | 48 |
| STACI-FFNG (Conf.) | NA | NA | NA | 0.948 | **1.850** | **1.555** | 391 |
| Deep RF | 0.733 | 0.175 | 0.452 | **0.954** | 5.433 | 2.264 | 103 |
| Deep GP | 0.633 | 0.191 | 0.356 | 0.941 | 4.781 | 2.788 | 70 |
| GPSat | 0.653 | **-0.064** | **0.287** | 0.958 | 3.476 | 2.272 | 2745 |
| SVGP | 0.690 | 0.276 | 0.395 | 0.956 | 5.150 | 3.237 | **20** |

The difference in prediction quality and UQ between the models can be visualized in Figure 2. The first row of Figure 2 represents the ground truth AOD values and the predicted means from the STACI Bayesian and conformal variants as well as the competitor models. The color scheme for this row goes from blue (lower values of opacity) to red (higher values of opacity). The second row represents the widths of the prediction intervals from each of the methods assuming a desired 95% coverage. The darker purple represents narrower widths that moves lighter into yellow which represents the wider. Comparing to the ground truth, we see STACI is able to capture more of the high pollution area than the competitors in mean surface prediction. Meanwhile, for the interval width representing UQ, we see that the conformal variant of STACI is the only one that has highly varied interval widths, while the UQ from most other methods is fairly stagnant in the middle of purple and yellow. The UQ benefit of GPSat is shown here where the local GPs provide personalized UQ similar to the conformal variant of STACI. The high variance areas from conformal and GPSat also correspond to areas with deep red high spikes in pollution which is commonly smoothed over in these classes of models.

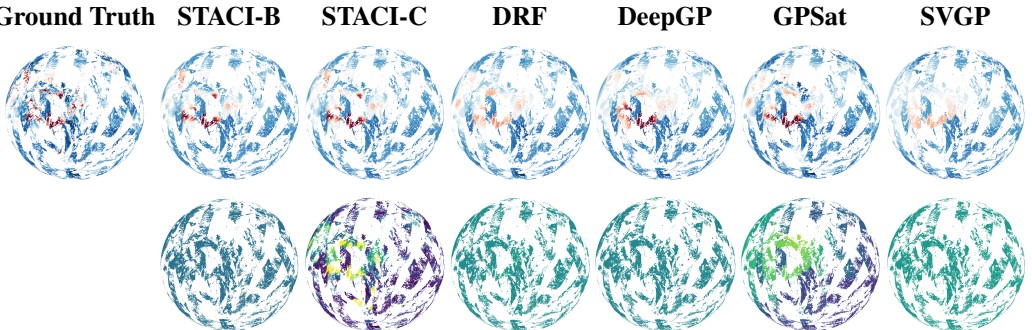

Figure 2: Predicted AOD surface values. Top row: predicted surface. Red indicates higher AOD values.; Bottom row: interval widths for Bayesian and conformal (STACI-C) uncertainty on AOD data. Darker shades denote narrower intervals.

## 4.3 Ablation Studies

Here, we perform two ablation studies for the AOD dataset to quantify model robustness. The first ablation study is impact of latent model dimension size on estimation error. The second ablation is impact of sampling percentage in training set construction on estimation error.

**Ablation Study: Latent dimension**
Figure 3a shows how the RMSE of each latent model varies as we change the latent dimension size from [32, 64, 128, 256]. We see that the two FFN models are fairly stable at around 0.58 for positional encoding and around 0.67 for Gaussian encoding. ResMLP error sees a noticeable decrease increasing from latent dimension 128 to 256. This indicates the FFN latent model results are stable to choice of latent dimension and the reported results are optimal.

**Ablation Study: Sampling Percentage**
Figure 3b shows how the RMSE of each approximate GP model varies as we change the training set sampling percentage at each time-step from [5%, 10%, 25%]. For STACI, we use the FFNP latent model. We see generally a decrease in estimation error with greater sampling percentage. Interestingly, DRF error rises slightly at the highest sampling percentage, but remains below the 5% case. We speculate this is due to fixing the noise parameter during Bayesian optimization. STACI still achieves the lowest error across settings.

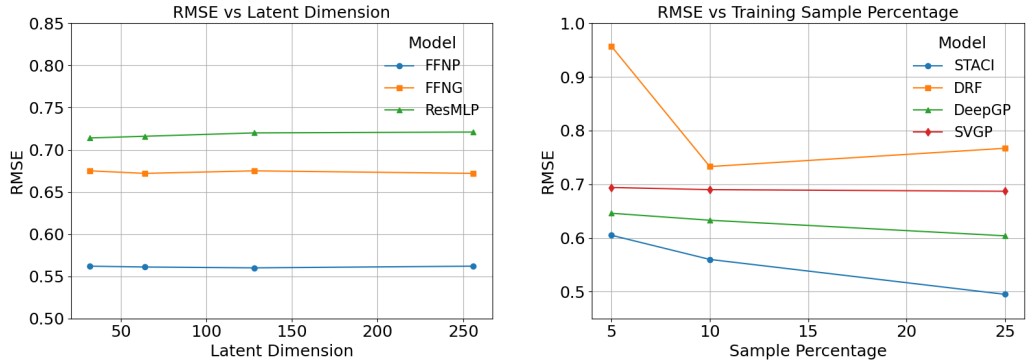

(a) RMSE vs. latent dimension size for FFNP, FFNG, and ResMLP.

(b) RMSE vs. train set sampling percentage for STACI, DRF, DeepGP, and SVGP.

Figure 3: Ablation Studies for STACI

# 5    Conclusion

In this work, we introduce the STACI algorithm, an approximate STGP combining high scalability and prediction accuracy with valid UQ. Compared to other deep and neural network based models, STACI is able to provide interpretable covariance parameters from simpler stationary kernels using the latent dimension expansion, along with the ability to recover the data ST correlation structure. A limitation of the approach is the computational expense in both speed and space of the SVGD training algorithm with larger numbers of models. With smaller model number, we are not able to fully explore the posterior space, and may converge to suboptimal estimates. This impacts the strong assumption of exchangeability of selected neighbors is impacted by our STACI's ability to accurately estimate the relevant covariance parameters. Additionally, the conformal algorithm as constructed is fairly simplistic, assigning equal importance to each selected neighbor. This can be optimized further with weighting proportional to ST distance. Nevertheless, STACI provides scalable aleatoric UQ and can be used in a variety of ST interpolation tasks.

One possible alternative direction is to use data-driven basis functions, such as the leading terms in a Karhunen-Loève expansion, as opposed to trigonometric functions in the output layer. We also believe that there are multiple possible extensions to non-Gaussian data. For example, adding a link function, or slightly changing the likelihood, should extend to distributions such as Bernoulli, or Poisson to model spatio-temporal correlated binary and count data. We also believe an extension to model spatial point processes is a feasible adaptation of the model. Following the work of [54], the Poisson process intensity function can be modeled as the square of a GP. This would allow continued utilization of both random Fourier features and our introduced approximation for efficient likelihood computation. These extensions would allow STACI to reach a wider audience and cover additional types of data.

## Acknowledgments

This work was supported by the National Institutes of Health (R01ES031651-01), the National Science Foundation (DMS2152887), and the U.S. Department of Energy (DOE), Office of Science (SC), Advanced Scientific Computing Research program under award DE-SC-0012704 and KJ0401010/CC147 and used resources of the National Energy Research Scientific Computing Center, a DOE Office of Science User Facility using NERSC award NERSC DDR-ERCAP0030592.

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
