# OpenReview forum: "STACI: Spatio-Temporal Aleatoric Conformal Inference"
_NeurIPS.cc/2025/Conference — NeurIPS 2025 poster_

### Official Review · Reviewer_AwKK · 2025-06-24

**Clarity:** 3
**Significance:** 3
**Originality:** 4
**Rating:** 5
**Confidence:** 4

**Summary:**

This paper proposes a novel methodology for estimating non-stationary random fields using feature construction, BNNs, and conformal predictions. The methods is well validated on two good examples and properly compared to SOTA methods in the field with promising results.

**Questions:**

- How would one perform model selection in this framework; as opposed to marginal likelihood maximization in GPs? Could this be elaborated on?

- The proposed method is, in some parts, complex. Which parts of the model could be simplified if the inference task at hand is perhaps simpler (for example not none-stationary)?

- What is the relation between (4) and Karhunen-Loeve expansions? If there is one, could this link be expanded on to help readers more familiar with these tools understand the proposed method?

**Ethical Concerns:**

["NO or VERY MINOR ethics concerns only"]

**Final Justification:**

Thank you to the authors for their time and effort. I believe this is a strong, well executed paper that solve an important problem in an effective and principled manner. I suggest acceptance.

**Limitations:**

Visual explanations of the use of conformal predictions could be helpful for readers less familiar with this class of methods.

**Paper Formatting Concerns:**

No Formatting Concerns

**Quality:**

4

**Strengths And Weaknesses:**

Strengths:
- The paper is well written, clear, and properly exposited. Shortcomings of other methods, and the proposed method, are well explained.
- The explanation on conformal predictions is succinct and clear.
- The numerics are well done and compare to many relevant methods.
- Overall the methods is quite original and tackles a challenging problem in spatio-temporal inference.


Weaknesses:
- In the numerics, the metrics for assessing performance are poorly explained, i.e.: Coverage, Interval, Score, Interval Width.
 Could these be elaborated on and explained how they are computed as they would help the reader.
-SVGD might suffer from mode-collapse in high dimensions, would this have a negative effect on the proposed methodology at test time on harder problems?

---

> ### Author Rebuttal · Authors · 2025-07-31
>
> We thank reviewer AwKK for taking the time to read through our work and provide valuable comments. We appreciate the praise for the clarity of our paper and originality of the methodology. We provide some rebuttals for individual comments here.
>
> > In the numerics, the metrics for assessing performance are poorly explained, i.e.: Coverage, Interval, Score, Interval Width. Could these be elaborated on and explained how they are computed as they would help the reader.
>
> We agree that descriptions of these metrics are helpful for readers and elaborated on these metrics further in the Supplementary Material Section B.3.
>
> > SVGD might suffer from mode-collapse in high dimensions, would this have a negative effect on the proposed methodology at test time on harder problems?
>
>    You raise a great point on the dangers of mode-collapse for difficult high-dimensional problems. While we have not found this to be a problem for our tested datasets, we acknowledge this could happen in other use cases and would affect the UQ from the Bayesian variant of STACI. However, we believe the conformal addition would still provide valid intervals as the exchangeability calculations are based on the mean estimates of the covariance parameters and not their predictive variances.
>
> > How would one perform model selection in this framework; as opposed to marginal likelihood maximization in GPs? Could this be elaborated on?
>
>   Model selection would be done through cross validation in our methodology for the model fitting and conformal hyperparameters. Priors for the covariance parameters used in the kernel function can also be adjusted for a more direct connection to marginal likelihood maximization in GPs.
>
> > The proposed method is, in some parts, complex. Which parts of the model could be simplified if the inference task at hand is perhaps simpler (for example not none-stationary)?
>
> We agree that this can be a complex framework. In a simpler stationary case, the need for the INR backbone is eliminated and this becomes a single hidden layer Bayesian neural network for the first half of our framework. Removal of the temporal component further reduces the number of priors set and model parameters needed to be learned.
>
> > What is the relation between (4) and Karhunen-Loeve expansions? If there is one, could this link be expanded on to help readers more familiar with these tools understand the proposed method?
>
> The KL transformation is an alternative to the Fourier transformation used in the spectral approximation. Both represent a stochastic process as a linear combination of basis functions. The KL transformation is a generalization of the principle components analysis (PCA) where the basis functions are orthonormal and chosen to maximize variance explained. A Fourier transformation restricts the basis functions to be trigonometric functions. Like PCA, application of the KL decomposition typically requires replication of the process to estimate the basis functions, whereas the Fourier approximation can be applied to a single realization of the space-time process by assuming a stationary covariance (in latent space, in our case).

---

> > ### Comment · Reviewer_AwKK · 2025-08-04
> >
> > Thank you to the authors for answering my queries about the paper. I believe adding this information to the paper will help readers less familiar with this topic. I will keep my score as is. Many thanks.

---

### Official Review · Reviewer_w1e9 · 2025-06-29

**Clarity:** 2
**Significance:** 3
**Originality:** 3
**Rating:** 5
**Confidence:** 3

**Summary:**

The problem with Gaussian Process approximations is that they only work well in low data cases. In this work the authors merge a Bayesian neural network approximation with the newly emerged conformal inference algorithm. This allows them to both produce a good approximation of the spatio-temporal data whilst still being able to accurately predict uncertainty quantification. The authors compare their work (in terms of RMSE and negative Gaussian log likelihood) with three common approaches from the literature on two datasets.

**Questions:**

- Please provide a clear description of figure 1.

- Please ensure all concepts discussed in the paper are clearly defined before used.

- Figure 2 is quite hard to interpret. Please provide more guidance.

- Further consideration for the conclusions would help. Especially what to do next.

**Ethical Concerns:**

["NO or VERY MINOR ethics concerns only"]

**Final Justification:**

Having read the reviews of others and the responses to those reviews I see that my original review was on the generous side. However, the comments to my questions lead me to believe that my score is now fitting for the paper. I do feel that the comments by WrPZ about missing baselines is a good one and strongly encourage the authors to achieve this for the final paper - this shouldn't just be a best effort.

**Limitations:**

yes

**Quality:**

3

**Strengths And Weaknesses:**

The paper is technically sound and covers the problem well. However, the clarity of the presentation could be improved to make the paper more approachable for those not closely working in this area. For example, there is a figure which provides an overview of the whole process, but this is not discussed or explained in the text; the concept of nugget variance is discussed but not defined.

---

> ### Author Rebuttal · Authors · 2025-07-31
>
> We appreciate reviewer w1e9 for taking the time to understand and provide feedback for our method. We appreciate the praise for soundness of our work and apologize for some areas that lack clarity. We provide some rebuttals for individual comments here.
>
> > Please provide a clear description of figure 1.
>
>   We apologize for lack of clarity and will add the following. Figure 1: STACI algorithm pipeline. Data from coordinates across space and time are first fed into our approximate spatio-temporal GP architecture and trained using SVGD. This allows for estimation of the latent space and prediction of the mean surface through the variational posterior. A conformal inference step is then fit to provide customized prediction intervals for valid uncertainty quantification.
>
> > Please ensure all concepts discussed in the paper are clearly defined before used.
>
> We apologize for not defining the nugget variance beforehand. The nugget variance is the portion of the variance that is fully random and can't be explained by our spatio-temporal model. This can be thought of as the random/measurement error term, or in other words the amount of aleatoric uncertainty. We will add this into the paper for the camera-ready version.
>
> > Figure 2 is quite hard to interpret. Please provide more guidance.
>
> Thank you for pointing this out!  We will add a more informative description akin to:
>
> The first row of Figure 2 represents the ground truth AOD values and the predicted means from the STACI Bayesian and conformal variants as well as the competitor models. The color scheme for this row goes from blue (lower values of opacity) to red (higher values of opacity). The second row represents the widths of the prediction intervals from each of the methods assuming a desired 95\% coverage. The darker purple represents narrower widths that moves lighter into yellow which represents the wider. We can see that the conformal variant of STACI is the only one that has highly varied interval widths, while the UQ from other methods is fairly stagnant in the middle of purple and yellow. The high variance areas from conformal also correspond to areas with deep red high spikes in pollution which is commonly smoothed over in these classes of models.
>
> > Further consideration for the conclusions would help. Especially what to do next.
>
> We thank the reviewer for highlighting this and further steps are important to extend this work. We believe that there are multiple possible extensions to non-Gaussian data. For example, adding a link function, or slightly changing the likelihood, should extend to distributions such as Bernoulli, or Poisson to model spatio-temporal correlated binary and count data. We also believe an extension to model spatial point processes is a feasible adaptation of the model. Following the work of Lloyd et al. (2015), the Poisson intensity can be modeled as the square of a GP. This would allow us to continue utilizing random Fourier features for efficient likelihood computation. These extensions would allow STACI to reach a wider audience and cover additional types of data.
>
>
> **Reference**
> Lloyd, Chris, et al. "Variational inference for Gaussian process modulated Poisson processes." International Conference on Machine Learning. PMLR, 2015.

---

### Official Review · Reviewer_WrPZ · 2025-07-02

**Clarity:** 3
**Significance:** 3
**Originality:** 3
**Rating:** 4
**Confidence:** 3

**Summary:**

This paper proposes STACI, a novel two-stage framework that combines a Bayesian neural approximation of non-stationary spatio-temporal GP with a conformal inference algorithm for uncertainty quantification. The authors address three key challenges in spatio-temporal modeling: (1) scalability, (2) non-stationarity, and (3) valid uncertainty quantification. The model leverages spectral approximations and a dimension expansion technique to approximate non-stationary covariance functions. STACI also incorporates conformal prediction to correct potential under/over-confidence introduced by approximations.

Empirical evaluations on both synthetic and real-world datasets demonstrate that STACI achieves superior prediction accuracy and tighter, valid prediction intervals compared to deep GPs, sparse GPs, and Deep RF methods.

**Questions:**

1.	While the proposed spectral approximation is theoretically justified, it would be helpful to understand the error bounds associated with the overall approximation. Specifically, how do factors such as the number of random features or SVGD particles affect the fidelity of the STGP approximation and the resulting prediction intervals?

2.	Have you tried to compare with the mixture model of local GPs in GPSat library?

3.	Why not using Continuous Ranked Probability Score(CRPS) to evaluate the quality of predicted uncertainties?

**Ethical Concerns:**

["NO or VERY MINOR ethics concerns only"]

**Final Justification:**

I appreciate the authors' thoughtful rebuttal and their efforts to incorporate feedback. The inclusion of CRPS results and the discussion on the spectral approximation and SVGD behavior address several of my concerns. However, two key points remain partially or fully unaddressed: (1) the lack of empirical comparison to standard DNN-based models, despite the method’s deep learning components, and (2) the omission of a known strong baseline (GPSat), which the authors acknowledged but have not yet evaluated. For these reasons, I have chosen to retain my original score.

**Limitations:**

yes

**Quality:**

3

**Strengths And Weaknesses:**

Strength:

1.	The authors propose a novel, scalable method that provides valid uncertainty quantification for non-stationary spatio-temporal processes—an important and under-explored area of research.

2.	The paper is well-written, and the methodology and preliminaries are clearly presented, making complex concepts easy to follow.

3.	The authors provide thorough experimental details, including ablations, training procedures, and model architecture. The visualization and results presented are clear and informative.

Weakness:

1.	Although the authors compared the proposed model with several GP-based models in the experiments, they did not include a comparison with DNN-based models.

2.	There are some typos and glitches here and there. E.g., Eq 9. Duplicate “at” in line 269.

---

> ### Author Rebuttal · Authors · 2025-07-31
>
> We'd like to thank the reviewer WrPZ for providing valuable comments for our work and directions for further improvement. We appreciate highlighting the novelty of our approach and clarity of the paper format. For the typos, we appreciate the reviewer for pointing these out and will fix these for the camera-ready version. We provide some rebuttals for individual comments here.
>
> > Although the authors compared the proposed model with several GP-based models in the experiments, they did not include a comparison with DNN-based models.
>
>  While we did not include DNN-based models, the backbone models for our latent space are popular Implicit Neural Representation DNN-based models currently in use for coordinate-based data. DNN-based models typically do not provide inherent uncertainty quantification, which was a major focus of our work.
>
> > While the proposed spectral approximation is theoretically justified, it would be helpful to understand the error bounds associated with the overall approximation. Specifically, how do factors such as the number of random features or SVGD particles affect the fidelity of the STGP approximation and the resulting prediction intervals?
>
> We agree that theoretical justification is important for practical use of this method and show that the spectral approximation is centered on the Matérn covariance function. The approximation accuracy increases with $J$, the number of basis functions, or the width of the final hidden layer in STACI. In Appendix B.4, we also show that increasing the number of SVGD particles does not have much of an effect on RMSE, but slighty decreases the NLL, indicating slightly better variance estimation at the cost of higher computational time.
>
> > Have you tried to compare with the mixture model of local GPs in GPSat library?
>
> While we did not compare with the mixture model from GPSat in our original study, we thank the reviewer for bringing it to our attention. From the Deep RF paper, we see it performs very well in this area, placing near the top in each metric across multiple datasets and showing more flexibility than traditional GPs. Unfortunately we could not implement it in time thus far, however we will make an effort to have results for the camera-ready version.
>
> > Why not using Continuous Ranked Probability Score (CRPS) to evaluate the quality of predicted uncertainties?
>
> Great point to use CRPS as a UQ quality metric! We include the CRPS results here for both datasets with the non-conformal variant of STACI. We see that STACI with FFNP backbone is still the highest performer across these use cases.
>
> For the MSS Data:
>
> | Model          | Mean CRPS |
> | -------------- | --------- |
> | STACI (FFNP)   | 0.0978    |
> | STACI (FFNG)   | 0.1327    |
> | STACI (ResMLP) | 0.2442    |
> | DRF            | 0.1062    |
> | DeepGP         | 0.1449    |
> | SVGP           | 0.2361    |
>
> AOD Data:
>
> | Model          | Mean CRPS |
> | -------------- | --------- |
> | STACI (FFNP)   | 0.2904    |
> | STACI (FFNG)   | 0.3685    |
> | STACI (ResMLP) | 0.3768    |
> | DRF            | 0.4522    |
> | DeepGP         | 0.3557    |
> | SVGP           | 0.3950    |

---

> ### Comment · Reviewer_WrPZ · 2025-08-07
>
> I appreciate the authors’ thoughtful rebuttal and their efforts to incorporate the feedback. My comments are as follows:
>
> While the authors make a reasonable point regarding the lack of inherent uncertainty quantification in DNN-based models, this does not fully justify the omission of standard DNN baselines for predictive performance comparison.
>
> The suggested comparison to the mixture model from GPSat has not yet been addressed with empirical results. Although it is understandable that implementation constraints exist, the absence of this comparison leaves a noticeable gap in the current evaluation.
>
> Given these outstanding concerns, I would prefer to retain my score as it currently stands.

---

### Official Review · Reviewer_8XoG · 2025-07-04

**Clarity:** 3
**Significance:** 1
**Originality:** 1
**Rating:** 3
**Confidence:** 3

**Summary:**

The paper consider the problem of inferring GP models using a variational Bayesian neural network approximation. A conformal inference algorithm is proposed and compared with three baselines including SVGP, deep RF, and deep GP.

**Questions:**

- How does the algorithm compare with baselines mentioned above, both theoretically and numerically?
- Can any guarantees be provided for the approximation methods used in this algorithm?
- Can any theoretical analysis on the improvement be provided?

**Ethical Concerns:**

["NO or VERY MINOR ethics concerns only"]

**Final Justification:**

The paper considers GP and not more complex problems like non-Guassian or Gaussian Cox processes. Based on the responses provided by the authors, I'm raising the review score.

**Limitations:**

yes

**Quality:**

2

**Strengths And Weaknesses:**

+ The idea of using a variational Bayesian neural network to approximate non-stationary spatio-temporal GPs makes sense.
+ The proposed solution is clearly described.
- The evaluation is quite limited and missed many important baselines, including both GP and more advanced Gaussian Cox process models, such as variational Bayesian approximation, mean-field approximation, path integral approximations of GP, and change of kernel in a reproducing kernel Hilbert space. Without a complete comparison, the benefit of the proposed method is hardly convincing.
- The pros and cons of the proposed algorithm compared to these baselines isn't clear.
- Can any guarantees be provided for the approximation methods used in this algorithm? Can any theoretical analysis on the improvement be provided?

---

> ### Author Rebuttal · Authors · 2025-07-31
>
> We would like to thank the reviewer 8XoG for taking the time to read through our paper and providing much constructive feedback. We appreciate the reviewer's acknowledgment of the clarity of our work and the logic of the method. Rebuttals to specific comments are provided.
>
> >The evaluation is quite limited and missed many important baselines, including both
> GP and more advanced Gaussian Cox process models, such as variational Bayesian
> approximation, mean-field approximation, path integral approximations of GP, and
> change of kernel in a reproducing kernel Hilbert space. Without a complete comparison, the benefit of the proposed method is hardly convincing.
>
> While we may have been unclear, we believe we have compared our method to many of these approaches. It is computationally infeasible to fit a GP to the full dataset, and all of the methods we compare to are approximations to a GP. We covered the variational Bayesian baseline by including SVGP. This KL-divergence based method was also shown to have better performance than the mean-field approximation in the original SVGP paper, so we did not include this in our comparisons. We also believe the Gaussian Cox process model, while advanced, is not in the scope of this work as we are not measuring count data across space and time. However, a potential area of future work is extending the method to non-Gaussian data, like the Gaussian Cox process model does. Replacing the Gaussian likelihood with other distributions such as Bernoulli or Poisson to handle geo-referenced binary or count data should be fairly straightforward. It should also be possible to extend the method to more challenging cases such as inhomogeneous Poisson processes for spatial point pattern data. For example, the model of Lloyd et al (2015) takes the Poisson process intensity function as the square of a GP. Thus the likelihood with random Fourier features approximating the GP can be computed analytically in O(n) flops, permitting the same computational approach taken here to fit the non-stationary Fourier model to spatial point pattern data.
>
> >The pros and cons of the proposed algorithm compared to these baselines isn’t clear.
>
> We explain the drawbacks of current baselines in paragraphs 2-4 of the introduction and apologize if this wasn't clear. Our method provides greater interpretability of covariance parameters than other deep GP methods and provides greater flexibility than the traditional GP. The conformal addition of our method is able to provide guaranteed coverage properties for prediction intervals, assuming the exchangeability condition holds. The drawback to the STACI method is the computational cost required for both SVGD training and conformal inference, however we do not believe it is prohibitive for most use cases.
>
> > Can any guarantees be provided for the approximation methods used in this algorithm? Can any theoretical analysis on the improvement be provided?
>
> We agree that theoretical guarantees need to be made for the approximation and provided a theorem on the spectral covariance approximation in supplement section A. We show that the spectral approximation is centered on the Matérn covariance function with accuracy that increases with $J$, the number of basis functions, or the width of the final hidden layer in our method.
>
> **Reference**
> Lloyd, Chris, et al. "Variational inference for Gaussian process modulated Poisson processes." International Conference on Machine Learning. PMLR, 2015.

---

> > ### Comment · Reviewer_8XoG · 2025-08-05
> >
> > I had some misunderstanding of the paper. Thanks to the authors for explaining the focus on GP, and not more complex problems like non-Gaussian processes or Gaussian Cox process. I'm raising my score after re-evlauating the paper.
> >
> > In terms of the guarantee, Theorem 1 as I understand only shows that the accuracy increases with the number of base function, which is quite straightforward. Can some guarantee be provided on the approximation gap?

---

### Decision · Program_Chairs · 2025-09-17

**Decision:**

Accept (poster)

**Comment:**

Three out of four reviewers gave very positive evaluations of this submission. Reviewer WrPZ raised a valid concern regarding an important missing baseline, which should be added in the final version. Based on the above, I recommend acceptance of this work.